# RadGraph: Extracting Clinical Entities and Relations from Radiology Reports

**Saahil Jain**[*]
Stanford University
saahil.jain@cs.stanford.edu

**Ashwin Agrawal**[*]
Stanford University
ashwin15@stanford.edu

**Adriel Saporta**[*]
Stanford University
asaporta@cs.stanford.edu

**Steven QH Truong**
VinBrain, VinUniversity

**Du Nguyen Duong**
VinBrain

**Tan Bui**
VinBrain

**Pierre Chambon**
Stanford University

**Yuhao Zhang**
Stanford University

**Matthew P. Lungren**
Stanford University

**Andrew Y. Ng**
Stanford University

**Curtis P. Langlotz**[†]
Stanford University
langlotz@stanford.edu

**Pranav Rajpurkar**[†]
Harvard University
pranav_rajpurkar@hms.harvard.edu

## Abstract

Extracting structured clinical information from free-text radiology reports can enable the use of radiology report information for a variety of critical healthcare applications. In our work, we present RadGraph, a dataset of entities and relations in full-text chest X-ray radiology reports based on a novel information extraction schema we designed to structure radiology reports. We release a development dataset, which contains board-certified radiologist annotations for 500 radiology reports from the MIMIC-CXR dataset (14,579 entities and 10,889 relations), and a test dataset, which contains two independent sets of board-certified radiologist annotations for 100 radiology reports split equally across the MIMIC-CXR and CheXpert datasets. Using these datasets, we train and test a deep learning model, RadGraph Benchmark, that achieves a micro F1 of 0.82 and 0.73 on relation extraction on the MIMIC-CXR and CheXpert test sets respectively. Additionally, we release an inference dataset, which contains annotations automatically generated by RadGraph Benchmark across 220,763 MIMIC-CXR reports (around 6 million entities and 4 million relations) and 500 CheXpert reports (13,783 entities and 9,908 relations) with mappings to associated chest radiographs. Our freely available dataset can facilitate a wide range of research in medical natural language processing, as well as computer vision and multi-modal learning when linked to chest radiographs.

## 1 Introduction

Radiology reports are comprised of free text containing critical information about a patient's health based on an interpretation of radiology images and clinical history. However, their unstructured nature combined with the complexity and ambiguity of natural language pose a challenge when using

---

[*]Equal Contribution
[†]Equal Contribution

35th Conference on Neural Information Processing Systems (NeurIPS 2021) Track on Datasets and Benchmarks.

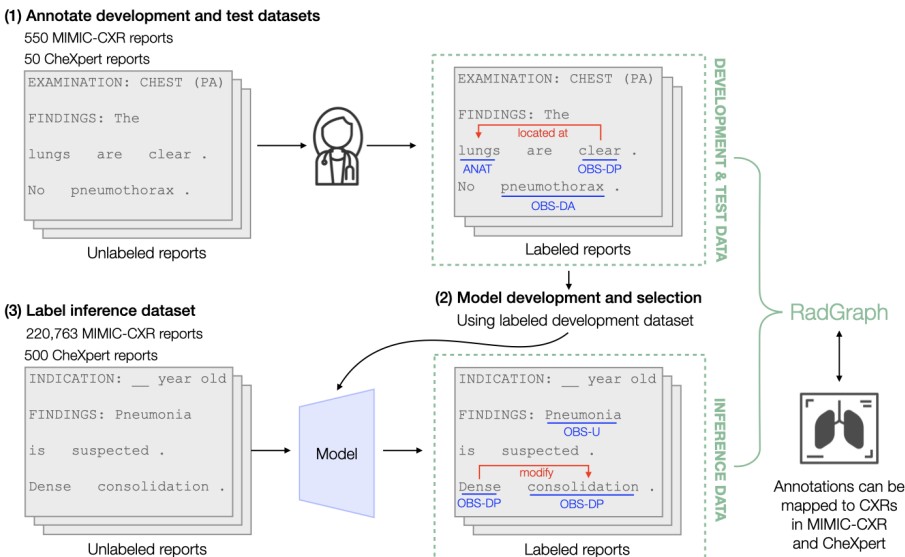

Figure 1: Process for developing RadGraph, our dataset of annotated entities and relations in radiology reports. First, board-certified radiologists annotate reports for our development and test datasets. Second, we train a deep learning model using the development dataset. Third, we use our model to automatically generate annotations for a much larger number of MIMIC-CXR and CheXpert reports, which can be mapped to associated chest radiographs.

radiology reports for clinical research and other downstream applications, especially in settings with limited labeled data. Automatically extracting clinically relevant information from radiology reports can enable various use cases, ranging from large-scale training of medical imaging models to disease surveillance.

Numerous approaches for extracting information from radiology reports have been developed with varying intended use cases. Large-scale datasets, such as MIMIC-CXR [1] and CheXpert [2], use automated radiology report labelers [2–6] to extract common medical conditions from reports. Other approaches [7–11] aim to extract more fine-grained information in reports. The development of automated approaches for structuring large amounts of clinically relevant information in reports is primarily limited by two factors. First, the choice of information extraction schema, such as the 14 medical conditions proposed by Irvin et al. [2], limits the amount of information extracted from reports. Second, there is a limited number of datasets with dense report annotations, which are expensive to obtain given the amount of time and expertise required by medical experts to procure such annotations.

In our work, we seek to address these limitations by creating RadGraph, a dataset of clinical entity and relation annotations for radiology reports. Our four primary contributions are the following. (1) We define a novel information extraction schema for radiology reports, intended to cover most clinically relevant information within the report while allowing for ease and consistency during annotation. (2) We release development and test datasets annotated according to our schema by board-certified radiologists. Our development dataset contains annotations for 500 radiology reports from the MIMIC-CXR dataset, consisting of 14,579 entities and 10,889 relations. Our test dataset contains two sets of independent annotations for 100 radiology reports from the MIMIC-CXR and CheXpert datasets. (3) We use our dataset to benchmark various modeling approaches. Our best approach, which we call RadGraph Benchmark, achieves a micro F1 of 0.94/0.91 (MIMIC-CXR/CheXpert) on named entity recognition and a micro F1 of 0.82/0.73 (MIMIC-CXR/CheXpert) on relation extraction. (4) We release an inference dataset, which contains annotations automatically generated by RadGraph Benchmark for 220,763 MIMIC-CXR reports, consisting of over 6 million entities and 4 million relations, and 500 CheXpert reports, consisting of 13,783 entities and 9,908 relations. Annotated reports in the inference dataset have mappings to associated chest radiographs, which can facilitate the development of multi-modal approaches in radiology. We summarize our process for creating RadGraph in Figure 1.

## 2 Related work

Various natural language processing (NLP) approaches have been developed and used to extract information from radiology reports. One approach uses automated radiology report labelers [2–6] to label reports within large chest radiograph datasets, such as MIMIC-CXR [1] and CheXpert [2], for a restricted number of common medical conditions. However, these labels do not capture critical fine-grained information contained in a radiology report, such as specific entities and their relations. While analysis tools [12–14] have been developed to extract key clinical concepts and their attributes from biomedical text and convert them into a structured format, such dictionary- and rule-based annotation systems are often limited in their report coverage and generalizability across institutions. Another approach aims to capture more specific and detailed information from radiology reports by adopting entity extraction schemas [8, 11] or schemas that focus on facts [10] and spatial relations [9, 15]. A central limitation of this approach is that it requires task-specific datasets to be densely annotated by domain experts.

To address this need for more specific annotations for radiology reports, datasets have been developed for information extraction from radiology reports. RadCore [8] is a multi-institutional database of radiology reports that contains entity-level annotations. PadChest [16] consists of chest radiographs and reports labeled with 174 different radiographic findings, 19 differential diagnoses, and 104 anatomic locations. Datta et al. [17] released a dataset of radiology reports annotated according to a schema based on spatial role labeling. Our dataset of radiology reports with dense annotations for both entities and relations extracts a broader range of information from the radiology text using a new information extraction schema designed for report coverage and generalizability. Outside the domain of radiology, several datasets have been specifically developed for the tasks of entity and relation extraction, such as SciERC [18] and SemEval 2017 Task 10 [19] for scientific information extraction.

Along with these datasets, there have been many advancements in NLP for the task of entity and relation extraction. Pipeline approaches [20, 21] decompose the task into separately trained subtasks (named entity recognition [22, 23] and relation extraction [24]), while joint extraction approaches [18, 25–27] model the two subtasks at the same time in order to capture interactions between entities and relations. When developing benchmarks for our task, we use both a pipeline approach [20] and a joint extraction approach [26].

## 3 Information extraction schema

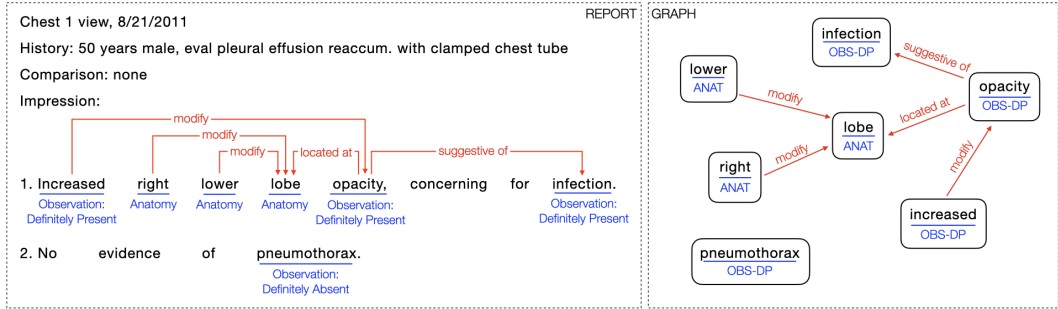

Figure 2: Sample report annotated according to the RadGraph schema (left) and the associated knowledge graph (right).

We propose a novel information extraction schema for extracting entities and relations from radiology reports, adapting the schema initially proposed by Langlotz et al. [7] to incorporate relations between entities and reduce the number of entities. After iterating on the initial schema based on feedback received from board-certified radiologists during labeling pilots, we design a schema that achieves two goals. First, our schema is designed for high coverage of the clinically relevant information in a report corresponding to the radiology image being examined, generally included in the Findings and Impression sections of the radiology report; we quantify the schema coverage in practice in Section 6.1. Second, our schema is designed to simplify the annotation task for radiologists, which improves labeling consistency and speed. Our schema is not designed to capture clinical context and history in

other report sections, as they contain information that is better documented elsewhere in the patient record and would require a more complex schema that would increase the difficulty of annotating.

**Entities**   We define an entity as a continuous span of text that can include one or more adjacent words. Entities in our schema center around two concepts: Anatomy and Observation. We specify three uncertainty levels for Observation, so our schema defines four entities: Anatomy, Observation: Definitely Present, Observation: Uncertain, and Observation: Definitely Absent. Anatomy refers to an anatomical body part that occurs in a radiology report, such as a "lung". Observations refer to words associated with visual features, identifiable pathophysiologic processes, or diagnostic disease classifications. For example, an Observation could be "effusion" or more general phrases like "increased".

**Relations**   We define a relation as a directed edge between two entities. Our schema uses three relations: Suggestive Of, Located At, and Modify. We select these three relations based on feedback from board-certified radiologists during labeling pilots to sufficiently cover the clinically relevant relationships between our entities with a relatively small number of relation types. We define the permitted use of each relation using the format: relation type (first entity type, second entity type). Suggestive Of (Observation, Observation) is a relation between two Observation entities indicating that the presence of the second Observation is inferred from that of the first Observation. Located At (Observation, Anatomy) is a relation between an Observation entity and an Anatomy entity indicating that the Observation is related to the Anatomy. While Located At often refers to location, it can also be used to describe other relations between an Observation and an Anatomy. Modify (Observation, Observation) or (Anatomy, Anatomy) is a relation between two Observation entities or two Anatomy entities indicating that the first entity modifies the scope of, or quantifies the degree of, the second entity. As a result, all Observation modifiers are annotated as Observation entities, and all Anatomy modifiers are annotated as Anatomy entities for simplicity. By using relations to indicate modification instead of distinct entity types for modifiers, which was the approach proposed by Langlotz et al. [7] and used by Hassanpour et al. [8], our schema can more precisely capture the modifying relationships while also simplifying the task for annotators. The distinction between modifiers and entities is often not semantically relevant or clear without relations, such as when modifiers modify other modifiers. Figure 2 contains an example of a report annotated according to our schema and the resulting graph.

## 4   Dataset

To facilitate the development of models that can extract information from radiology reports according to our schema, we release RadGraph, a dataset containing radiology reports along with annotated entities and relations for each report. Our dataset includes development and test datasets, which were annotated by board-certified radiologists, as well as an inference dataset, which was annotated by our benchmark deep learning model.

### 4.1   Development and test datasets

Table 1: Development and test dataset annotation statistics

|  | Train (%) | Dev (%) | Test - MIMIC-CXR (%) | Test - CheXpert (%) |
|---|---|---|---|---|
| Anatomy | 5,366 (43.3) | 987 (45.0) | 528 (40.8) | 640 (43.4) |
| Observation: Definitely Present | 5,046 (40.7) | 831 (37.9) | 478 (37.0) | 603 (40.9) |
| Observation: Uncertain | 587 (4.7) | 93 (4.2) | 43 (3.3) | 55.5 (3.8) |
| Observation: Definitely Absent | 1,389 (11.2) | 280 (12.8) | 244 (18.9) | 175 (11.9) |
| Total Entities | 12,388 (100) | 2,191 (100) | 1,293 (100) | 1,473.5 (100) |
| Modify | 5,641 (61.0) | 1,013 (61.8) | 522 (57.8) | 696 (62.9) |
| Located at | 3,179 (34.4) | 554 (33.8) | 354 (39.2) | 348 (31.5) |
| Suggestive of | 431 (4.7) | 71 (4.3) | 26.5 (2.9) | 62.5 (5.6) |
| Total Relations | 9,251 (100) | 1,638 (100) | 902.5 (100) | 1,106.5 (100) |

To construct our development and test datasets, we obtained radiology report annotations according to our schema from three board-certified radiologists, each with at least eight years of experience. We ran three labeling pilots, which included around 15 reports each, to train our radiologists and

Table 2: Development, test, and inference dataset demographic statistics

| | Attribute | Train (%) | Dev (%) | Test MIMIC-CXR (%) | Test CheXpert (%) | Inference MIMIC-CXR (%) | Inference CheXpert (%) |
|---|---|---|---|---|---|---|---|
| Sex | Male | 54.6 | 72.0 | 10.0 | 48.0 | 51.2 | 54.4 |
| | Female | 45.4 | 28.0 | 90.0 | 52.0 | 48.6 | 45.6 |
| | Unknown | 0.0 | 0.0 | 0.0 | 0.0 | 0.2 | 0.0 |
| Age | 0-20 | 1.9 | 1.3 | 4.0 | 2.0 | 0.9 | 0.6 |
| | 21-40 | 5.9 | 6.7 | 0.0 | 22.0 | 11.5 | 13.8 |
| | 41-60 | 26.8 | 21.3 | 16.0 | 36.0 | 28.4 | 31.2 |
| | 61-80 | 40.9 | 69.3 | 66.0 | 32.0 | 41.3 | 46.6 |
| | 81+ | 24.5 | 1.3 | 14.0 | 8.0 | 17.7 | 7.8 |
| | Unknown | 0.0 | 0.0 | 0.0 | 0.0 | 0.2 | 0.0 |
| Race | White | 65.4 | 56.0 | 32.0 | 46.0 | 61.2 | 41.8 |
| | Black | 16.0 | 20.0 | 62.0 | 4.0 | 15.2 | 4.6 |
| | Hispanic | 4.0 | 0.0 | 0.0 | 8.0 | 5.2 | 27.2 |
| | Asian | 2.6 | 0.0 | 0.0 | 8.0 | 3.2 | 7.0 |
| | Native | 0.5 | 0.0 | 0.0 | 0.0 | 0.3 | 1.4 |
| | Other | 4.0 | 21.3 | 0.0 | 18.0 | 4.4 | 8.4 |
| | Unknown | 7.5 | 2.7 | 6.0 | 16.0 | 10.4 | 9.6 |

iteratively improved our schema based on their feedback. To ensure high quality annotations, we do not include any of the annotations obtained during pilot labeling initiatives in our released dataset. The radiologists used a text labeling platform (Datasaur.ai [28], Sunnyvale, CA) to directly annotate the free-text reports according to our schema. We provide a breakdown of our development and test datasets by entities, relations, and data splits in Table 1. Additionally, in Table 2 we provide demographic breakdowns by sex, age, and race for our development and test datasets, as well as for our inference dataset, which we describe in Section 4.2.

**Development dataset**   We sample 500 radiology reports from the MIMIC-CXR dataset [1] for our development dataset. Each report is annotated by a single board-certified radiologist, resulting in 14,579 entities and 10,889 relations across all reports. Our development dataset was divided into train and dev sets, where the dev set includes 15% of the development dataset. Patients associated with reports in the train set and dev set do not overlap.

**Test dataset**   We sample 50 radiology reports from the MIMIC-CXR dataset and 50 radiology reports from the CheXpert dataset [2] for our test dataset in order to test generalization of approaches across institutions. Each report is independently annotated by two board-certified radiologists, resulting in an average of 2,766 entities and 2,009 relations per annotator across all reports. Patients associated with reports in the MIMIC-CXR test dataset and the development dataset do not overlap.

## 4.2   Inference dataset

First, we develop and measure the performance of a deep learning model called RadGraph Benchmark, which we describe further in Section 5, using our development and test datasets respectively. Next, we use Radgraph Benchmark to automatically annotate 220,763 reports from the MIMIC-CXR dataset and 500 reports from the CheXpert dataset.

On our selected MIMIC-CXR reports, we annotate 6,161,934 entities (2,699,288 Anatomy, 2,386,057 Observation: Definitely Present, 273,301 Observation: Uncertain, 803,288 Observation: Definitely Absent) and 4,409,026 relations (2,683,628 Modify, 1,554,406 Located At, 170,992 Suggestive Of). On our selected CheXpert reports, we annotate 13,783 entities (5,825 Anatomy, 6,453 Observation: Definitely Present, 515 Observation: Uncertain, 990 Observation: Definitely Absent) and 9,908 relations (6,467 Modify, 3,084 Located At, 357 Suggestive Of).

## 4.3   De-identification of reports

Each report in our dataset is de-identified according to the US Health Insurance Portability Act (HIPAA). MIMIC-CXR reports have already been de-identified by Johnson et al. [1], who replaced

protected health information (PHI) in reports with three consecutive underscores. We de-identify CheXpert reports using an automated, transformer-based de-identification algorithm followed by manual review of each report. PHI is replaced with fake PHI following a hiding-in-plain-sight (HIPS) [29] approach. The de-identification of the CheXpert reports was confirmed by manual review.

## 4.4 Data usage and ethics

Our dataset with documentation and associated code is hosted and maintained on PhysioNet under the following license: PhysioNet Credentialed Health Data License 1.5.0. It can be accessed at the following link: `https://doi.org/10.13026/hm87-5p47`.

Our data can be used for various purposes in the healthcare domain. We highlight two use cases. One use case is to develop NLP models for entity and relation extraction in radiology using our development dataset. Another use case is to develop multi-modal models in radiology using our inference dataset, which enables linkage of full-text radiology reports, knowledge graphs (entities/relations as per our schema), and associated chest radiographs from the MIMIC-CXR and CheXpert datasets. We also release the checkpoint for our model, RadGraph Benchmark, which can automatically annotate radiology reports. Models developed using our dataset can have clinical impact in various ways. Examples can range from population-level analysis using entities and relations automatically extracted from radiology reports to AI-assisted diagnosis using medical imaging models that can automatically generate knowledge graphs from radiology images.

To avoid any potential harm to patients, researchers training models on our datasets should take into account potential distribution shifts that may occur when they apply their models to other datasets with different patient populations, as discussed further in Section 6.2. As recommended by Seyyed-Kalantari et al. [30], when deploying clinical models in practice, researchers should audit performance disparities across attributes, such as sex, age, and race, which we report in Table 2 for our datasets.

## 4.5 Limitations

First, as mentioned in Section 3, our schema does not capture clinical context in radiology reports, such as information included in the Comparison or History sections. Second, as discussed in Section 6, there exist ambiguous cases in radiology reports that can be difficult to label according to our schema. Third, our annotations are limited to chest X-ray radiology reports from the MIMIC-CXR and CheXpert datasets, although the RadGraph schema is designed to annotate radiology reports in general. Fourth, the reports for both the MIMIC-CXR and CheXpert datasets are collected from hospitals only in the United States (Beth Israel Deaconess Medical Center in Boston, MA, and Stanford Hospital in Stanford, CA, respectively). Fifth, as discussed in Section 6, our test dataset is independently labeled by two radiologists, with more inter-observer variability on the CheXpert test set than on the MIMIC-CXR test set. Sixth, our datasets are not consistently balanced across demographic attributes. We report demographic statistics across all reports in our development, test, and inference datasets in Table 2.

## 5 Benchmarks

### 5.1 Approaches

We propose an entity and relation extraction task for radiology reports that can be developed using our development dataset and tested using our test dataset. To support the development of methods for our task, our dataset processes each radiology report into a sequence of space-delimited tokens, where punctuation like commas and semicolons have been separated from words to support entity recognition. For each report, we provide annotations identifying the type and span of each entity as well as relations between entities. We describe several initial approaches to entity and relation extraction for our task below.

**Baseline model**   We develop a simple, transformer-based Baseline approach. Our Baseline approach to entity and relation extraction uses a BERT [31] model with a linear classification head on top of the last layer for NER and R-BERT [32] for relation extraction. For our baseline NER approach, since

Table 3: Performance on relation extraction by approach

| | MIMIC-CXR | | CheXpert | |
|---|---|---|---|---|
| | Micro F1 | Macro F1 | Micro F1 | Macro F1 |
| Radiologist Benchmark | 0.947 | 0.910 | 0.745 | 0.704 |
| *Baseline* | | | | |
|    BERT | 0.468 | 0.372 | 0.424 | 0.356 |
|    BioBERT | 0.507 | 0.412 | 0.451 | 0.387 |
|    Bio+Clinical BERT | 0.454 | 0.367 | 0.389 | 0.343 |
|    PubMedBERT | 0.436 | 0.356 | 0.385 | 0.335 |
|    BlueBERT | 0.428 | 0.339 | 0.341 | 0.282 |
| *DYGIE++* | | | | |
|    BERT | 0.805 | 0.752 | 0.712 | 0.688 |
|    BioBERT | 0.801 | 0.731 | 0.701 | 0.668 |
|    Bio+Clinical BERT | 0.806 | 0.739 | 0.701 | 0.672 |
|    PubMedBERT | **0.823** | **0.783** | 0.725 | **0.692** |
|    BlueBERT | 0.803 | 0.712 | 0.705 | 0.664 |
| *PURE* | | | | |
|    BERT | 0.805 | 0.731 | 0.722 | 0.648 |
|    BioBERT | 0.806 | 0.757 | 0.721 | 0.654 |
|    Bio+Clinical BERT | 0.809 | 0.746 | 0.728 | 0.664 |
|    PubMedBERT | 0.812 | 0.745 | **0.729** | 0.679 |
|    BlueBERT | 0.818 | 0.738 | 0.699 | 0.655 |

the same entity may span multiple tokens, we use the IOB tagging scheme [33] and convert IOB tags to entity types defined by our schema after inference. We use a learning rate of 2e-5 (tuning range 2e-4 to 2e-6) with a batch size of 8 for our BERT-base model for NER after hyper-parameter tuning, consistent with the tuning approach used by Devlin et al. [31], and a learning rate of 2e-5 with a batch size of 4 with 16 gradient accumulation steps for our R-BERT model for relation extraction, consistent with the approach used by Wu et al. [32] except we use a smaller batch size.

**Benchmark models**     We develop additional benchmark approaches for our task, using two different entity and relation extraction model architectures. Our first approach uses the DYGIE++ framework by Wadden et al. [26], which achieved state-of-the-art at the time on NER and relation extraction by jointly extracting entities and relations. Our second approach uses the Princeton University Relation Extraction system (PURE) by Zhong et al. [34], which achieved state-of-the-art at the time on relation extraction using a pipeline approach that decomposes NER and relation extraction into separate subtasks.

We use BERT in both our PURE and DYGIE++ approaches. For PURE NER, we select a learning rate of 1e-5 (tuning range 1e-4 to 1e-6) with a batch size of 16 for BERT and a learning rate of 5e-5 (tuning range 5e-3 to 5e-5) for task specific layers. For PURE relation extraction, we use a learning rate of 2e-5 (tuning range 2e-4 to 2e-6) with a batch size of 16 for BERT. For PURE NER, we use a span length of three for the PURE approach since only 0.3% of entities in our development dataset consist of more than three words. For PURE relation extraction, we use a context window of 50 words based on average sentence length in our train set. For DYGIE++, we use a learning rate of 5e-5 with a batch size of 1 for BERT and a learning rate of 1e-3 for task specific layers, consistent with the approach used by Wadden et al. [26].

**Biomedical pretraining**     For each of our approaches, in addition to using BERT weight initializations, we use weight initializations from four different biomedical pretrained models, which are BioBERT [35], Bio+ClinicalBERT [36], PubMedBERT [37], and BlueBERT [38].

**Training details**     We train our models using NVIDIA GeForce GTX 1080 GPUs (1 for Baseline, 1 for DYGIE++, and 3 for PURE) until convergence, once for each approach before testing. Time to convergence varies across approaches, taking $\sim$ 1 hour for Baseline NER, $\sim$ 5 hours for Baseline relation extraction, $\sim$ 2 hours for DYGIE++ (joint NER and relation extraction), $\sim$ 1 hour for PURE NER, and $\sim$ 10 hours for PURE relation extraction.

Table 4: Benchmark performance on entity recognition

| | Anatomy | Observation: Definitely Present | Observation: Uncertain | Observation: Definitely Absent | Micro F1 | Macro F1 |
|---|---|---|---|---|---|---|
| | MIMIC-CXR | | | | | |
| Radiologist Benchmark | 0.994 | 0.981 | 0.953 | 0.996 | 0.988 | 0.981 |
| RadGraph Benchmark | 0.968 | 0.922 | 0.700 | 0.952 | 0.940 | 0.886 |
| | CheXpert | | | | | |
| Radiologist Benchmark | 0.944 | 0.917 | 0.757 | 0.960 | 0.928 | 0.894 |
| RadGraph Benchmark | 0.941 | 0.884 | 0.714 | 0.910 | 0.905 | 0.862 |

Table 5: Benchmark performance on relation extraction

| | Modify | Located At | Suggestive Of | Micro F1 | Macro F1 |
|---|---|---|---|---|---|
| | MIMIC-CXR | | | | |
| Radiologist Benchmark | 0.952 | 0.949 | 0.830 | 0.947 | 0.910 |
| RadGraph Benchmark | 0.804 | 0.861 | 0.685 | 0.823 | 0.783 |
| | CheXpert | | | | |
| Radiologist Benchmark | 0.741 | 0.779 | 0.592 | 0.745 | 0.704 |
| RadGraph Benchmark | 0.709 | 0.779 | 0.588 | 0.725 | 0.692 |

## 5.2 Evaluation metrics

We report both micro and macro F1 for entity recognition and relation extraction. For entity recognition, a predicted entity is considered correct if the predicted span boundaries and predicted entity type are both correct. For relation extraction, a predicted relation is considered correct if the predicted entity pair is correct (both the span boundaries and entity type) and the relation type is correct. For a radiologist benchmark, we compute metrics using the two independent sets of radiologist annotations collected on our test set. For modeling approaches, we compute two metrics, where each metric uses annotations acquired by a different labeler as ground truth, and then average the metrics. We report results on the MIMIC-CXR and CheXpert test sets separately.

## 5.3 Results

First, we compare the performance of each approach developed using our development dataset alongside our radiologist benchmark on both the MIMIC-CXR and CheXpert test sets. When comparing approaches, we use the strict relation extraction metric defined above as the primary end-to-end approach metric, as it uses both predicted entities and relations in its computation. Both DYGIE++ and PURE approaches obtain higher micro and macro F1 scores than the Baseline approach but lower micro and macro F1 scores than the radiologist benchmark on both test sets. We find that the DYGIE++ approach with PubMedBERT initializations achieves the highest micro and macro F1 scores on the MIMIC-CXR test set and the highest macro F1 score on the CheXpert test set for relation extraction. Accordingly, we call this approach the **RadGraph Benchmark**. We report our results for all approaches in Table 3.

Second, we specifically evaluate the performance of RadGraph Benchmark alongside the radiologist benchmark on both entity recognition and relation extraction. We report metrics on the MIMIC-CXR test set followed by the CheXpert test set for RadGraph Benchmark and the human benchmark as follows. RadGraph Benchmark achieves a micro F1 of 0.94/0.91 on named entity recognition and a micro F1 of 0.82/0.73 on relation extraction. The human benchmark achieves a micro F1 of 0.99/0.93 on named entity recognition and a micro F1 of 0.95/0.75 on relation extraction. Both RadGraph Benchmark and the human benchmark obtain lower micro F1 scores on the CheXpert test set compared to the MIMIC-CXR test set for both named entity recognition and relation extraction. For entity recognition, both benchmarks obtain the highest F1 scores on Anatomy and Observation: Definitely Absent and the lowest F1 score on Observation: Uncertain for both test sets. For relation extraction, both benchmarks obtain the lowest F1 scores on Suggestive Of for both test sets. We report the full results for the benchmarks across entity types in Table 4 and across relation types in Table 5.

Table 6: Schema coverage

| Average per report | Development | Test - MIMIC-CXR | Test - CheXpert |
|---|---|---|---|
| Sentences | 6.6 | 6.1 | 8.2 |
| Sentences annotated | 5.8 | 5.6 | 5.8 |
| Sentences annotated (%) | 87.7% | 92.3% | 70.7% |
| Tokens | 62.0 | 52.7 | 61.7 |
| Tokens annotated | 28.7 | 25.6 | 31.3 |
| Tokens annotated (%) | 46.4% | 48.6% | 50.8% |

## 6 Analysis

### 6.1 Schema coverage

Given that existing information extraction systems for radiology reports often suffer from a lack of report coverage [8], we measure the number of tokens and sentences in report sections covered by our schema. To calculate coverage, we extract the Findings and Impression sections of the reports, which our schema is designed to annotate. We then calculate the average percent of sentences and tokens annotated per report across the development and test datasets. For the token-level metrics, we exclude punctuation. We find that annotations obtained using our schema cover a high percentage of sentences across the development and test datasets. In the relevant sections of reports in the development dataset, an average of 87.7% of sentences and 46.4% of tokens were annotated per report. The token annotation percentage includes words that do not contain clinically relevant information, such as stop words. Although the schema coverage in the MIMIC-CXR test set resembles the coverage of the development dataset, in the relevant sections of reports in the CheXpert test set, an average of 70.7% of sentences and 50.8% of tokens were annotated per report, suggesting that information in CheXpert reports tends to be more concentrated in particular sentences. We report these results across the development and test sets in Table 6.

### 6.2 Annotation disagreements

We explore disagreements that occur when annotating according to our schema. To measure agreement between radiologists using our schema, we calculate Cohen's Kappa [39] between the two annotators on each test set for the named entity recognition task and the relation extraction task separately. For named entity recognition, we compute Kappa scores of 0.974 and 0.829 on the MIMIC-CXR and CheXpert test sets respectively. For relation extraction, we compute Kappa scores of 0.841 and 0.397 on the MIMIC-CXR and CheXpert test sets respectively. One reason for greater disagreement on the CheXpert test set compared to the MIMIC-CXR test set may relate to different percentages of patients in the intensive care unit (ICU) within the MIMIC-CXR dataset and the CheXpert dataset, which can systematically affect the contents of radiology reports. A second reason for greater disagreement may result from a higher concentration of annotations in a smaller number of sentences in the CheXpert test set, as reported in Section 6.1. Denser annotations can be more complicated to label, as they are more likely to contain layered relations that present ambiguities and difficulties described below.

We find five primary categories of inter-observer variability between annotators on our test dataset. First, we observe disagreements related to the direction of relations. For example, given the phrase "bands of coarse linear opacity," one annotator annotated "bands," "coarse," and "linear" modifying "opacity", while the other annotated "bands" modifying "coarse", "coarse" modifying "linear", and "linear" modifying "opacity". Second, we observe disagreements annotating implantable devices such as tubes, lines, and catheters. For example, for the phrase "pleural tube", one annotator annotated "pleural" as an Anatomy referencing the intended location of the tube. However, another annotator annotated "pleural" as an Observation, where "pleural" indicates the type of tube. We consider the second annotation to be correct, given that "pleural tube" does not indicate an anatomic location, particularly when misplaced. Third, given that we only provide three levels of uncertainty, we observe disagreements annotating certain Observations as Definitely Present or as Uncertain, such as "distal tip" in the phrase "with distal tip not clearly seen". Fourth, we observe disagreements related to annotating a phrase as a single large entity or multiple smaller entities. For example, one radiologist annotated "cephalad portion" as a single entity, while the other annotated it as two separate entities; in general, we instruct radiologists to provide more granular annotations where possible. Fifth, we

observe disagreements for challenging cases for which our schema does not mandate one correct answer, such as the phrase "right greater than left pleural effusions".

Next, we explore disagreements between RadGraph Benchmark and radiologist annotators. We find that the five areas of annotator disagreement described above likewise explain many of these disagreements. For example, while a radiologist annotator annotated the phrase "with suggestion of osteopenia" as Observation: Definitely Present, the model annotated the phrase as Observation: Uncertain; this example falls under our third category, in which the level of uncertainty for an Observation may be ambiguous. Additionally, we find that RadGraph Benchmark incorrectly disagrees with radiologist annotators when annotating rarer words in radiology reports, such as "fat pad".

## 7 Conclusion

In our work, we introduce RadGraph, a dataset of clinical entities and relations annotated in full-text radiology reports using a novel information extraction schema for structuring radiology reports. First, we propose our schema, which is designed to extract clinically relevant information associated with a radiologist's interpretation of a medical image in a radiology report. Second, we release development and test datasets annotated by board-certified radiologists. Each report is densely annotated, resulting in 14,579 entities and 10,889 relations extracted from 500 reports in our development dataset. Third, we develop a deep learning model, called RadGraph Benchmark, which obtains a micro F1 of 0.82/0.73 (MIMIC-CXR/CheXpert) on relation extraction. Fourth, we release an inference dataset annotated by our model for 220,763 MIMIC-CXR reports and 500 CheXpert reports, extracting around 6 million entities/4 million relations and 13,783 entities/9,908 relations from each set of reports respectively. Each report in the inference dataset can be linked to its associated chest radiograph in the MIMIC-CXR or CheXpert dataset.

By proposing a new schema and dataset for extracting clinically relevant information from unstructured text, we hope that RadGraph can facilitate a wide range of research in natural language processing, as well as computer vision and multi-modal learning, in various medical domains.

## 8 Acknowledgements

We would like to acknowledge Datasaur.ai for generously providing us access to their labeling platform. We would like to acknowledge Leo Anthony Celi and Tom Pollard from the MIMIC-CXR team and Nigam Shah, Ethan Chi, and Omar Khattab from Stanford University for their support.

Research reported in this publication was made possible in part by the *National Institute of Biomedical Imaging and Bioengineering (NIBIB)* of the National Institutes of Health under contracts 75N92020C00008 and 75N92020C00021.

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
