# OpenReview forum: "RadGraph: Extracting Clinical Entities and Relations from Radiology Reports"
_NeurIPS.cc/2021/Track/Datasets_and_Benchmarks/Round1 — NeurIPS 2021 Datasets and Benchmarks Track (Round 1)_

### Official Review · Reviewer_yUeJ · 2021-06-30
**Review of the paper titled "RadGraph: Extracting Clinical Entities and Relations from Radiology Reports The "**

**Rating:** 9
**Confidence:** 3
**Correctness:** I do not have any concerns with the c…
**Clarity:** The paper is well written

**Strengths:**

1. The paper is very well-written and a joy to read.
2. The paper enriches existing datasets of through annotations that are trained on a smaller radiologist annotated data.
3. I like the fact that the authors clearly list out the limitations of their data annotation procedure in Section 4.5
4. Section 6.2 is insightful. It clearly explains disagreements even among radiologists and intricacies of Mimic-CXR vs. CheXpert on annotation difficulty.

**Weaknesses:**

1. A discussion and empirical demonstration of how this annotated dataset can used for downstream healthcare tasks would have been nice. While the current case-studies are based on machine learning, it is not clear how doctors would use the dataset.

2. The authors use three relations of Suggestive Of, Located At, and Modify. Why did the authors limit themselves to only these three? Are these three exhaustive enough to cover all possible relationships in the dataset? Were these relationships decided by radiologists?



**Additional Feedback:**

All my feedback is outlined in the strengths, weaknesses, and Documentation fields.

**Documentation:**

The website clearly describes the dataset schema.

The authors should have clearly mentioned that the access to the dataset is available in the supplementary package. Otherwise, it requires the reviewer to log in, which may reveal identity.

I would appreciate if the authors share the codebase of their studies. Reproducing the results just from the given information in the paper would be hard.

**Ethics:**

I do not have any concerns

**Relation To Prior Work:**

The paper clearly contextualizes the work and contributions made over existing work/datasets.

**Summary And Contributions:**

The paper contributes an entity-relation graph of radiology reports from chest XRays. The annotations on a subset of this dataset were made by certified radiologists. This subset is used to train a neural model and annotate the rest of the dataset. Accuracy of the neural model is analyzed and is found to have an F1 in the range [0.73,0.82].

---

> ### Author Response · Authors · 2021-07-13
> **Thank you for the thoughtful feedback and questions, which we attempt to answer below!**
>
> We thank the reviewer for reading our paper, providing encouraging comments, and asking thoughtful questions.
>
> To address the reviewer’s point about better communicating the downstream usefulness of our dataset, in our latest revision uploaded, we expand the scope of Section 4.4 to provide more concrete examples of how our dataset can have clinical impact. Regarding the reviewer’s question about how doctors might benefit from our dataset, one such example, which we’ve added to the paper, would be a more comprehensive AI-assisted diagnosis tool that can generate knowledge graphs from chest X-rays. Our inference dataset could ideally facilitate the development of such a tool to improve the workflow of doctors. While existing automated chest X-ray diagnosis tools primarily output the presence or absence of particular medical conditions, such a tool could offer another, unique value proposition to medical providers.
>
> Next, we appreciate the reviewer’s question about the selection of the three relations in our schema. As we now mention in our latest uploaded revision in Section 3, we include these three relations in our schema based on feedback from board-certified radiologists to comprehensively cover the clinically relevant relationships between our entities with a relatively small number of relation types. Since one of the core advantages of our schema is its simplicity, which improves labeling consistency and speed, we purposefully focus on a small but comprehensive set of relations, which were suggested and verified by board-certified radiologists. Our three, broad relations (Suggestive Of, Modify, and Located At) cover each of the possible directed relations between our entities, except for the relation from Anatomy to Observation. Given the context of radiology reports, we purposefully exclude this relation, as the Anatomy should be the root of a relation with an Observation. For example, for the phrase “lobe opacity”, the “opacity” (Observation) describes the “lobe” (Anatomy); we do not find clinically relevant examples of relations in the opposite direction, which is why we do not include this direction for simplicity.
>
> Regarding the reviewer’s comment about anonymously accessing the dataset, the reviewer can use the special URL with the password in the URL section (below the Supplementary package download link) to access the dataset without needing to log in. After entering the URL / password, code associated with our paper can then be viewed. This way, the reviewer can access our code for replication, which they may not have been able to view without the above information.
>
> We appreciate the reviewer’s questions and suggestions to improve our paper’s clarity!

---

### Official Review · Reviewer_hUFJ · 2021-07-05
**This paper presents a new dataset RadGraph that consists of entities and relations extracted from medial test. Benchmark models are training on this dataset an released for external use. These dataset and models will be very useful to the community.**

**Rating:** 9
**Confidence:** 4
**Clarity:** The paper is well written and easy to…

**Strengths:**

* This paper presents a new dataset of entities and relations in full-chest X-ray radiology reports based on a new information extraction schema designed to structure radiology reports. The released datasets are annotated by board-certified radiologists. The authors also present baseline models trained on these datasets. These dataset could be important  to further research in information extraction from text for both clinicians and researchers.
* The paper also describes and provides baseline models PURE and DYGIE++ approaches with BERT.
* The dataset and models are externally released

**Weaknesses:**

The paper is well written and checks all major boxes. below are some suggestions:
* Recently deep learning model in medical computing have shown to provide disparate performance across patient demographics. It would be great if the dataset is balanced across population groups and if not this is documented  and added as  a limitation.
* While the paper does a good job of explaining the technical aspects of the problem and the solution I think it could expand on the description of clinical significance.
* A broader impact section should be added


**Additional Feedback:**

I recommend adding a broader impact section detailing any limitations or ethical concerns when using this dataset.

**Correctness:**

* The development and test dataset is prepared by collecting annotations from two radiologist ensuring the correctness of the dataset
* The schema is also revised based on radiologist feedback. It is designed to contain high coverage of clinically relevant information and to simplify the annotation task

**Documentation:**

Dataset statistics is provided In Table 1 and described in corresponding section 4.

**Ethics:**

* The authors could include a broader impact section
* What is the accuracy across population demographics? Is the dataset balanced across demographics and if not what are the expected limitations of this dataset?

**Relation To Prior Work:**

Related work section provides adequate information on existing words. Existing labels do not capture fine-grained information such as entities and relations. Also, some works are limited in report coverage and generalizability. The dataset claims to improve upon these works which are later demonstrated via results.

**Summary And Contributions:**

1. The paper presents a new dataset RadGraph that consists is a dataset of entities and relation in full-text chest X-ray
    * train set consists of radiologist annotations for 500 radiology reports
    * test set consists two independent sets of radiologist annotations for 100 independent reports
2. The authors present a baseline deep learning model that achieves 0.82 and 0.73 F1 scores
3. Inference dataset generated by RadGraph benchmark model is released freely

---

> ### Author Response · Authors · 2021-07-13
> **Thank you for the encouraging feedback and helpful suggestions!**
>
> We thank the reviewer for reading our paper and providing helpful comments!
>
> Regarding the reviewer’s point about the demographics of our dataset, we agree that providing information and clarity regarding demographics is important. In our latest revision, we provide a demographic breakdown by sex, age, and race across our development, test, and inference datasets in Table 2 in Section 4. As suggested, we then add a limitation to Section 4.5, which acknowledges that our dataset is not consistently balanced across demographic attributes. Adding more data for underrepresented groups is an impactful area of future work.
>
> Next, to address the reviewer’s point regarding the inclusion of a broader impact section, we expand the scope of Section 4.4 to discuss the broader impact and significance of our dataset. In the latest uploaded revision, we provide more concrete examples of how our dataset can have clinical impact, and we add an additional ethical consideration related to measuring the performance across demographic attributes when deploying clinical models in practice. We found that including this information in Section 4.4 allows us to efficiently communicate the reviewer’s suggestions.

---

> > ### Comment · Reviewer_hUFJ · 2021-07-20
> > **Thanks or the revision!**
> >
> > Thanks for the detailed response and revised version of the paper. The revision with additional discussion on broader impact and significance section as well as addition of demographic information has further improved the utility of this dataset. I do not have additional concern.

---

### Official Review · Reviewer_wUJ3 · 2021-07-06
**Excellent contribution and dataset for automated annotation of radiology free text reports**

**Rating:** 9
**Confidence:** 4

**Strengths:**

Their novel information extraction schema is clinically relevant, logical, and addresses limitations in previous schemas. Their deep learning model is rigorously bench-marked with excellent performance. The datasets are collected with high-fidelity, well-documented, and freely available. The work is broadly relevant to the medical NLP and computer vision communities with the potential stimulate multiple studies and lines of research.

**Weaknesses:**

Not a weakness, per se, but inclusion of de-identified demographic metadata such as age or binned age (45-50), gender/sex, or ethnicity in future versions/releases would allow studies to examine whether RadGraph or other algorithms show similar performance across different demographic groups or whether hidden disparities exist.


Updated review:
The revised manuscript addresses this issue with the addition of Table 2.

**Additional Feedback:**

Very nice work, it was a pleasure to review! A couple of general comments for future releases of the dataset, if applicable. It could be useful if the metadata included some de-identified demographic information such as age or binned age (45-50), gender/sex, or ethnicity. The reason is that future studies may wish to test whether algorithms show similar performance across different demographic groups or whether there are hidden disparities. However, the reviewer understands that this information either may not be available since the work is partially derived from existing datasets or that the information may be available in the original datasets and that future studies can map to these metadata if needed. Also, the size and utility of the dataset could be enhanced in the future if there was the ability to crowd-source radiologist verification of the annotations in the inference dataset, which could then be incorporated in new dataset releases. The reviewer acknowledges this is not the intent of the work, and it would require significant effort, but it is worth considering for future releases.

**Clarity:**

Overall, the manuscript was well-written with a detailed description of the dataset and methods as well as compelling results. Manuscript formatting adheres to the guidelines, and no spelling or grammar errors were identified.

**Correctness:**

The dataset is rigorously constructed by expert radiologists after three labeling/training pilots. The submission includes a Datasheet, ML Reproducibility Checklist, supplement, and is hosted on a public data repository.

**Documentation:**

The dataset is very well documented, and I applaud the authors for their thorough submission. The data is hosted on PhysioNet for long-term hosting with an associated DOI and an appropriate license. The description of dataset curation and organization is clear with a detailed "dataset datasheet".

**Ethics:**

Ethical issues are discussed. The reviewer has no concerns.

Updated review:
The expanded discussion of ethical and broader implications in Section 4.4 is appreciated.

**Relation To Prior Work:**

This is a significant contribution that has the potential to assist the development of large scale, structured annotation of free-text radiology reports. Relevance to previous work is discussed.

**Summary And Contributions:**

Jain et al present a novel information extraction schema for free text radiology reports and apply this to existing datasets (MIMIC-CXR and CheXpert) to generate a new, curated dataset of structured clinical annotations. Next, they train a deep learning benchmark, RadGraph, to automatically extract clinical entities and relationships. They thoroughly document dataset curation and benchmark their deep learning model to show excellent performance at identification of clinical entities and relationships. Their densely annotated, expert curated development and test datasets are released along with an inference dataset with automatically generated annotations, which are all freely available on Physio.net. Overall, the work is very strong and a significant contribution to natural language processing for radiology report annotation. In addition, the datasets have potential to stimulate multiple lines of work in the NLP, computer vision, and multi-modal learning community.

Updated review:
The revised manuscript is strong and addresses my previous concerns.

---

> ### Author Response · Authors · 2021-07-13
> **Thank you for the kind feedback and helpful comments!**
>
> We would like to thank the reviewer for reading our paper and providing valuable ideas for additions to our paper.
>
> We acknowledge the reviewer’s feedback regarding the inclusion of de-identified demographic information like age, sex, and race. In our latest revision, we add these recommended statistics in Table 2 of our paper, which includes a demographic breakdown by sex, age, and race across our development, test, and inference datasets. We appreciate this helpful suggestion.
>
> We also agree that increasing the size and utility of the dataset by crowd-sourcing verification of the annotations in our dataset would be an interesting and impactful line of work, which we are excited to further investigate for a potential future release.

---

### Official Review · Reviewer_UuYD · 2021-07-08
**Review of "RadGraph: Extracting Clinical Entities and Relations from Radiology Reports"**

**Rating:** 9
**Confidence:** 3
**Clarity:** Written very well

**Strengths:**

- multiple datasets used
- association between full text radiology reports and imaging
- board-certified radiologist annotations used

**Weaknesses:**

- no validation in other countries; MIMIC and CheXpert are both USA

**Additional Feedback:**

Great work!

**Correctness:**

The claims seem correct; I would include some word on sample size limitations in the limitations section; this goes in line with the fact that datasets from the same country were used, likely from pretty similar populations given BIDMC and Stanford are in Boston and Palo Alto, respectively

**Documentation:**

Great

**Ethics:**

Great -- ethical use of data

**Relation To Prior Work:**

Builds upon prior work as detailed in the paper; the authors seem judicious about their claims but nonetheless push forward the technology in the field

**Summary And Contributions:**

The authors introduce RadGraph which associates full-text radiology reports with clinical entities that makes use of a novel information extraction scheme to structure radiology reports. They introduce and employ the schema and use a deep learning model, RadGraph Benchmark, with MIMIC-CXR and CheXpert datasets. This contribution is an important one in the field of natural language processing and computer vision with potential applications in a wide range of other fields in medicine.

---

> ### Author Response · Authors · 2021-07-13
> **Thank you for the encouraging feedback and suggestions!**
>
> We are very thankful to the reviewer for reading our paper and providing such encouraging feedback.
>
> We acknowledge the reviewer’s recommendation to expand our limitation section to comment on the limited number of countries reflected in our dataset. In our latest revision, we add the locations of the two hospitals reflected in our datasets in the fourth limitation in Section 4.5. We agree that expanding the dataset to reflect more patients across various countries would be an interesting area of future work, and we are excited to look more into the feasibility of such a project.

---

### Author Response · Authors · 2021-07-13
**Thank you! We have uploaded a revised version of our paper.**

We would like to thank each of the reviewers for their encouraging feedback and helpful comments. Based on this feedback, we have uploaded a revised version of our paper which includes the following additional information:
* We include demographic breakdowns by various attributes (sex, age, and race) across our development, test, and inference datasets in Table 2 (recommended by Reviewer wUJ3 and Reviewer hUFJ)
* We add two additional points to Section 4.5, which discusses the limitations of our dataset, related to the location of the hospitals associated with our datasets (recommended by Reviewer UuYD) and the demographics of our datasets (recommended by Reviewer hUFJ)
* We expand Section 4.4, which focuses on data usage and ethics, to include a discussion of the broader clinical impact of our datasets as well as an additional ethical consideration (recommended by Reviewer hUFJ and Reviewer yUeJ).
* We add a brief explanation of why we chose the relations in our schema in Section 3 (recommended by Reviewer yUeJ).

---

### Decision · Program_Chairs · 2021-07-26

**Decision:**

Accept

**Comment:**

Dear authors,

thank you very much for your submission and fruitful discussion with the reviewers.

The final score are 9/9/9/9 (with confidences 3/4/4/3) which is obviously a clear accept.

All reviewers agreed in the importance of the data, the work and scientific rigor that went into creating and annotating the data.

The only real limitation was the limitation of US data, which cannot be changed easily.